# The Application of Finite Element Method for Analysis of Cross-Wedge Rolling Processes—A Review

**DOI:** 10.3390/ma16134518

**Published:** 2023-06-21

**Authors:** Zbigniew Pater

**Affiliations:** Mechanical Faculty, Lublin University of Technology, 36 Nadbystrzycka Str., 20-618 Lublin, Poland; z.pater@pollub.pl

**Keywords:** cross-wedge rolling, FEM, damage, application

## Abstract

The aim of this article is to review the application of the finite element method (FEM) to cross-wedge rolling (CWR) modeling. CWR is a manufacturing process which is used to produce stepped axles and shafts as well as forged parts for further processing on forging presses. Although the concept of CWR was developed 140 years ago, it was not used in industry until after World War 2. This was due to the limitations connected with wedge tool design and the high costs of their construction. As a result, until the end of the twentieth century, CWR tools were constructed by rolling mill manufacturers as they employed engineers with the most considerable experience in CWR process design. The situation has only changed recently when FEM became widely used in CWR analysis. A vast number of theoretical studies have been carried out in recent years, and their findings are described in this overview article. This paper describes nine research areas in which FEM is effectively applied, namely: the states of stress and strain; force parameters; failure modes in CWR; material fracture; microstructure modeling; the formation of concavities on the workpiece ends; CWR formation of hollow parts; CWR formation of parts made of non-ferrous materials; and new CWR methods. Finally, to show the potential of FEM on CWR modeling, a CWR process for manufacturing a stepped shaft used in car gearboxes is simulated numerically. This numerical simulation example shows that FEM can be used to model very complex cases of CWR, which should lead to a growing interest in this advanced manufacturing technique in the future.

## 1. Introduction

Cross-wedge rolling (CWR) is an advanced manufacturing technique for producing stepped axles and shafts that are used in the automotive, machine-building and railway industry. This technique can also be employed to produce axisymmetric preforms for press forging [1].

Although the concept of CWR was already established in the 19th century, the method was not employed on a wider scale until the 1960s. This was due to the considerable complexity of CWR, which made the industrial application of this process difficult. Early studies on the theory and technology of CWR were based on costly experiments. A comprehensive and in-depth review of the state-of-the-art CWR processes was published in 1993 by Fu and Dean [2]. In the last decade of the 20th century, CWR began to be widely studied by theoretical methods, including the energy method and the simplified analysis method using the upper bound theorem. The use of these methods made it possible to estimate relatively quickly the force parameters in CWR and predict failure modes in CWR production, including workpiece necking (rupture) and uncontrolled slip [3,4,5].

The first studies on the application of the finite element method (FEM) to CWR modeling were published at the turn of the 20th and 21st centuries. The relatively late application of FEM to study this process is due to the fact that CWR can only be modeled in a 3D state of deformation, and that the stroke of the tools (wedges) exceeds by many times the dimensions of the workpiece, which results in a longer CPU time. Since then, the application of FEM to CWR modeling has become standard practice, and the use of FEM has led to increased technological capabilities and popularity of this interesting manufacturing technique.

Considering that FEM has been employed to CWR modeling for twenty-five years, this is a suitable occasion to summarize studies to date that have dealt with this issue. The author of this paper has undertaken to review previous studies on CWR modeling. This review article is divided into nine sections relating to the main research directions in this field. To illustrate the potential of FEM for CWR modeling, toward the end of the paper, a cross-wedge rolling process of a gearbox shaft is simulated numerically.

## 2. States of Stress and Strain

The use of FEM made it possible to examine changes in stresses in the workpiece during CWR. The first study on this problem was conducted in 2000 by Dong et al. [6]. A total of 14 rolling conditions were simulated by varying CWR process parameters. The simulations were conducted using the ANSYS/LS-DYNA program equipped with an explicit time integration scheme. The first principal stress *σ*_1_ and the maximum shear stress *τ_max_* at selected points of the workpiece were examined. It was found that *σ*_1_ > 0 at the center of the workpiece, which could lead to internal crack formation. In 2002, Fang et al. [7] simulated the CWR process conducted with two rigid material dies. The simulations were performed using the implicit FEM program Deform 3D. Theirs was the first study to show the maps of principal stress *σ*_1_ and mean (hydrostatic) stress *σ_m_* distribution in the cross-section of the workpiece.

In 2003, Pater [8] used MSC.MARC Autoforge to determine the distributions of the three principal stresses at five different points located in the cross-section of the workpiece that was deformed by three different rolling methods, i.e., with the use of flat rolls, two rolls, and three rolls. Based on the obtained values of *σ*_1_, *σ*_2_, *σ*_3_ (expressed in MPa), he determined the distributions of a stress state coefficient *WK*, given by:(1)$$WK=\frac{\sigma_{m}}{\sigma_{i}}\;\left[-\right],$$ where (2)$$\sigma_{m}=\frac{\sigma_{1}+\sigma_{2}+\sigma_{3}}{3}\;\left[\mathrm{MPa}\right],$$
(3)$$\sigma_{i}=\sqrt{{\left(\sigma_{1}-\sigma_{2}\right)}^{2}+{\left(\sigma_{1}-\sigma_{3}\right)}^{2}+{\left(\sigma_{2}-\sigma_{3}\right)}^{2}}\;\left[\mathrm{MPa}\right].$$

A comparison of the obtained *WK* values made it possible to establish a relationship between the basic parameters of CWR and internal crack formation.

Strain distributions in CWR-formed parts were first presented by Dong et al. [6] and had the form of diagrams plotted for selected points located in the cross-section of the workpiece. The first maps showing effective strain distributions in the cross-section of the workpiece were presented by Pater in a monograph [9] published in 2001. The numerical results confirmed that the effective strain distribution had a layered (annular) pattern and that the highest strain was located in the outer layer (near the surface of the workpiece). Effective stress and strain distributions determined for different parameters of CWR are given, e.g., in [1].

It is now standard practice to examine stresses, strains, and temperatures in workpieces deformed by CWR.

## 3. Force Parameters

The use of FEM made it possible to accurately predict force and energy parameters in CWR. Torque distributions were first presented by Fang et al. [7]. In turn, the first distributions of forming force (the load acting on the wedge) were given by Pater [8]. In the study, the numerical distributions were compared with experimentally obtained ones in order to verify their accuracy. This method of numerical model validation was also employed by Bartnicki and Pater [10] to study the CWR of hollow shafts. Given the ease of measuring force parameters, the above FEM model validation method became standard in the years that followed.

In 2007, Shu et al. [11] investigated the relationship between the basic parameters of CWR (i.e., forming angle *α*, spreading angle *β*, area reduction Δ*A*, initial billet diameter *d*_0_) and torque. The numerical results provided valuable insight in terms of CWR mill design. Peng and Zhang [12] used FEM to calculate the axial force stretching the workpiece and thus causing a change in its dimension. The axial force calculation is indispensable to predict whether necking (rupture) of the workpiece will occur. This problem was further investigated by Shu et al. [13], who determined the effect of rolling parameters on the axial force in CWR.

## 4. Failure Modes in CWR

One of the first problems investigated by FEM was the prediction of failure modes in CWR. These failure modes include uncontrolled slip, core necking (rupture), and internal crack formation in the workpiece.

In 1998, Dong et al. [14] investigated the problem of slip between the wedges and the workpiece. They developed a FEM model of CWR, which was then used in [15,16] to determine the effect of friction factor *μ*, forming angle *α*, and area reduction Δ*A* on the interfacial slip. Lovell [17] found that slip could be prevented by maximizing the value of friction coefficient describing the contact condition between the wedge and the workpiece. This could be achieved by making special serrations on the forming surfaces of the wedges. Urankar et al. [18] performed FEM simulations and found that the critical friction coefficient at which slip did not occur was two times higher for hollow parts than that for solid ones.

When the stresses induced by the axial component of the rolling force exceed the yield stresses, there may occur necking (rupture) of the workpiece core. The first numerical simulation showing a necking formation mechanism was performed by Pater in [19]. In a later study, Jia et al. [20] investigated the effect of the number of wedge passes on core necking formation. They found that the best way to prevent it was by ensuring the same cross-sectional reduction in two consecutive passes of the wedge tool.

In CWR processes for producing hollow parts, the workpiece may undergo crushing. Urankar et al. [21] showed that this defect was linked to billet wall thickness and area reduction. Another external defect that may occur in CWR is the excessive bending of the workpiece. A FEM analysis, which was conducted by Shu et al. [22], demonstrated that the defect was particularly likely to occur in multi-wedge CWR, in which the workpiece was deformed by several pairs of wedge-shaped tools simultaneously.

The above defects are easy to detect in the workpiece because their occurrence can be spotted with the naked eye. What poses a more serious problem is the internal crack formation due to the Mannesmann effect [1]. Since the problem of modeling material fracture in CWR has been investigated in many studies, the author of this paper has decided to discuss this issue in a separate section. In contrast, the analysis of failure modes causing the formation of external defects (e.g., bending, overlap, cross-sectional distortion, slip, necking) poses no major difficulties these days and is performed at the stage of design of new CWR techniques.

## 5. Material Fracture in CWR

The first study [23] investigating material fracture in CWR focused on the effective strain distribution in the axial zone of the workpiece. According to the authors of the study, this parameter was the optimum criterion for predicting internal crack formation. Nevertheless, later studies demonstrated that, for crack formation prediction, it was more convenient to employ energy-based fracture criteria. These criteria are used to determine the so-called damage function, which is given by the following equation:(4)$$f_{i}=\int_{0}^{\varepsilon }\Psi \left(\sigma \right)d\varepsilon ,$$
where *f_i_* is the damage function determined according to *i*-th criterion, Ψ(σ) is the stress state function, and *ε* is the effective strain. For material fracture to occur, the damage function *f_i_* must reach the critical value *C_i_*, which is determined via so-called calibration tests.

A modified version of the Cockcroft–Latham ductile fracture criterion was first used for material fracture analysis. Using this criterion and the Forge^®^ program, Piedrahita et al. [24] determined the effect of the basic parameters of CWR (*α*, *β*, and Δ*A*) on the value of the damage function *f_CL_* in the axial zone of the workpiece. The same program and the same criterion were employed by Silva et al. [25] to simulate the crack formation mechanism in CWR. The simulations were conducted with the killing element technique, which allowed for particular elements to be deleted when the critical value of the damage function was reached therein. In later studies, the normalized Cockcroft–Latham criterion was frequently employed in numerical investigations of CWR processes. For instance, Xia et al. [26] investigated the material fracture mechanism in warm CWR of 42CrMo steel shafts, while Bulzak [27] demonstrated that the likelihood of material fracture could be reduced when the CWR process was conducted with concave tools.

Other material fracture criteria have also been used in numerical analyses of CWR processes. Novella et al. [28] used the Oyane–Sato criterion to simulate the CWR of AA6082-T6 bars. Cakircali et al. [29] employed the Johnson–Cook failure model to investigate the CWR of Ti6Al4V parts. Attempts have also been made to determine the optimum criterion for modeling crack formation in CWR. Pater et al. [30] used nine different criteria to simulate the CWR process of a harrow tooth preform, but none of the criterion was capable of reproducing the real process with pinpoint accuracy. Bulzak [31] made use of nine damage criteria to investigate the CWR process of a railcar axle. The study showed that the accurate material fracture predictions were obtained with the criteria developed by Ayada, Brozzo et al., Ko et al., and Rice and Tracey. Pater et al. [32] employed 10 different damage criteria to predict the size of internal cracks in rolled parts. The best results were obtained using the criteria developed by Argon et al., Oyane, Freudenthal, and Brozzo et al. All these criteria were based on the effective stress *σ_i_*, which means that they took into account both normal and shear stresses for crack formation prediction. Bearing these findings in mind, Pater et al. [33] developed a new damage criterion wherein both the maximum principal stress *σ*_1_ and the maximum shear stress *τ_max_* were taken into consideration to predict material fracture. This criterion served as a basis for developing a method for rapid material fracture prediction in CWR [34], which made use of damage function maps obtained via an FEM analysis of 54 cases of CWR conducted with variable parameters. In addition to that, the new material fracture criterion was used in studies investigating the relationship between material fracture and the use of rolling method [35] and wedge-shaped tools with variable angles [36].

The accuracy of predicting material fracture in CWR depends not only on the applied damage function *f_i_*, but also on the critical damage function value *C_i_*. It is important that the stress state in a calibration test used for the determination of *C_i_* should reflect as much as possible the stress state in CWR. For that reason, research studies were conducted to develop new calibration tests that could be used for CWR analysis. The first tests were based on the rotary compression of a cylindrical specimen by flat plates [37] and rotary compression of a disc specimen in tool cavity [38,39], in which the state of stress was close to that in CWR. Next, a CWR-based test was developed [40], the use of which yielded excellent results with respect to material fracture prediction.

Material fracture in CWR can also be predicted based on stress analysis. Yang et al. [41] used the ANSYS program to simulate the effect of area reduction on the principal stresses in the center of the workpiece. The numerical results demonstrated that the stresses were the least conducive to internal crack formation when the area reduction Δ*A* was 55%. Zhou et al. [42] simulated several cases of CWR for varying values of *α*, *β*, and Δ*A*. The effect of these parameters on the mean stress was determined, demonstrating that the likelihood of material fracture increased when the CWR process was conducted with lower values of *α*, higher values of *β*, and a greater value of area reduction Δ*A*. Zhao et al. [43] studied the effect of the wedge tip fillet on the maximum principal stress *σ*_1_. The results showed that an increase in the tip fillet value caused an increase in *σ*_1_, which led to a higher likelihood of internal crack formation. Zhou et al. [44] performed the numerical analysis of CWR using Deform 3D, finding that the maximum shear stress *τ_max_* caused the formation of internal voids that would then develop into cracks due to the impact of the maximum principal stress *σ*_1_. Zhou et al. [45,46] proposed a new stress-based criterion that took into account the effect of both *τ_max_* and *σ*_1_. A total of 12 cases of CWR taken from the literature were modeled numerically by this criterion, showing that the criterion was an effective tool for predicting material fracture in CWR.

Despite numerous studies on internal crack formation in CWR, this problem has not yet been satisfactorily solved. It is believed that future research on material fracture in CWR will deal not only with the prediction of cracking, but also with the determination of its exact location and size.

## 6. Microstructure Modeling

Wang et al. [47] were the first to use FEM for predicting the mean grain size in a part produced by CWR. They used the Deform 3D program to investigate the CWR process for manufacturing parts made of the AISI5140 grade steel. In a subsequent study [48], they developed a numerical model involving dynamic and static recrystallization kinetics, and the results of the FEM model validation showed high agreement between the experimental and numerical mean grain sizes. In later studies, FEM was employed to determine the distributions of grain size in parts formed by CWR from the following materials: nickel-based superalloy GH4169 [49,50], steel grades 42CrMo [51] and 25CrMo4 [52,53], aluminum alloy grade 6061 [54], and titanium alloy grade Ti6Al4V [55]. In addition to that, Li et al. [56] determined the volume fraction of the α phase in TC6 alloy parts rolled with varying temperature, area reduction, and rotational speed.

A separate research problem relates to the numerical modeling of micro-cross-wedge rolling, in which micro-components (the diameter of the cylindrical workpiece was 0.8 mm) made of pure copper were cold-formed. Jiang et al. [57] used the ANSYS/LS-DYNA program to model the micro-cross-wedge rolling of workpieces with their grain size ranging from 6 to 240 μm. The numerical results showed that the critical grain size of the workpiece had to be 40 μm in order to obtain a uniform stress distribution. Wei et al. [58] used the same program to investigate the effect of geometrical and process parameters (*α*, *β*, Δ*A*) on the micro-cross-wedge rolling of 1100H16 alloy micro-components.

The prediction of mechanical properties and the microstructure of parts produced by CWR will be one of the main future research directions. It is expected that along with the development of microstructure models for new materials, the entire manufacturing processes will be simulated, including operations such as preheating, rolling, and heat treatment.

## 7. Formation of Concavities on the Workpiece Ends

When the billet material undergoes deformation on its ends, the material flows on the surface of the workpiece, which results in the formation of concavities on both ends of the workpiece. The defect should be removed by cutting off the defective ends of the workpiece. This, however, leads to material losses and thus reduces the cost-effectiveness of CWR. Numerous studies have been conducted to reduce material losses in CWR. Zeng et al. [59] used Deform 3D to that end, finding that this defect could be controlled by using profiled billets (with tapered or circular-arc ends). Guo and Lu [60] determined the effect of angles *α* and *β* and area reduction Δ*A* on the volume of defective ends. Wei et al. [61] proposed a new CWR technique to prevent this defect; nevertheless, the implementation of the solution required a significant elongation of the wedge tools. To reduce material losses resulting from concavity formation on the workpiece ends, Yang et al. [62] proposed the use of tapered billets, while Han et al. [63,64] considered using variable cone angle billets.

Pater et al. [65] investigated 21 cases of CWR in order to determine a relationship between the basic parameters of CWR and the depth of concavities. Equations were established for calculating the concavity allowance.

For the workpiece ends to be shaped in such a way that the concavity allowance is reduced, an additional forming operation must be performed. Wang et al. [66] proposed a new roll-cutting forming method of conical end blanks for cross-wedge rolling. The proposed solution was validated numerically via Deform 3D.

## 8. CWR Formation of Hollow Parts

The problem of forming hollow parts by CWR has been undertaken in many studies, which stemmed from the need to reduce material consumption in CWR. The first FEM analysis of CWR for producing a hollow part without using a mandrel was conducted in 2004 by Pater and Bartnicki [10]. The numerical results obtained by rolling with flat wedges showed agreement with the experimental findings. Pater et al. [67] used FEM to determine the effect of basic process parameters (*α*, *β*, Δ*A*) on the formation of cross-sectional ovalization. Additionally, the study was the first to simulate CWR conducted with the use of a mandrel for making a hole in the shaft. The numerical study conducted in 2005 by Bartnicki and Pater [68] demonstrated that CWR by three rolls was the best solution for forming hollow parts.

Sun et al. [69] investigated the forming of a hollow axle by synchrostep cross-wedge rolling conducted with the use of three pairs of wedges at the same time. A similar study, in which the workpiece was deformed simultaneously by five pairs of wedges, was conducted by Hu et al. [70]. The multi-wedge CWR of a hollow railcar axle was investigated by Peng et al. [71]. A similar process with the use of a mandrel was modeled by Zheng et al. [72]. In turn, Pater et al. [73] proposed an innovative CWR process for producing a hollow railcar axle by three coupled rolls.

Stepped tubes are another type of CWR-formed hollow part that have been modeled by FEM. Ma et al. [74] investigated the CWR of a rear axle tube, focusing on the effect of process parameters on the grain size and uniformity. The production of this part was also studied in [75]. The authors of the study focused on preventing the workpiece from undesirable axial displacement during rolling. This was achieved by adding a technological constraint step at the billet end. Yang and Hu [76] examined the problem of ovalization. in forming a hollow shaft by CWR with mandrel. The study showed that a decrease in *β* and an increase in *α* led to reduced cross-sectional ovalization.

Numerical analyses have also been conducted on the manufacture of hollow valves by CWR. Ji et al. [77] investigated the CWR of a hollow valve made of the 4Cr95Si2 grade steel. The numerical analysis made it possible to determine the values of the wedge angles, ensuring the stability of the rolling process. Ji et al. [78] analyzed the effect of CWR parameters on the inner bore in a hollow valve preform. To achieve a uniform bore diameter, a mandrel was used in CWR [79].

Many studies have focused on the role of mandrel in cross-wedge rolling hollow parts. Huang et al. [80] used Deform 3D to determine the effect of mandrel diameter on stresses and strains, material flow, and workpiece temperature in CWR. It was found that the diameter of the mandrel had impact on the cross-sectional reduction. Shen et al. [81] studied CWR conducted using a detachable mandrel with variable diameters. Zhou et al. [82] proposed a modified CWR roll design with a curved-surface knife. This solution proved effective in preventing the hole diameter increase at the location where the wedge tools would cut into the workpiece.

Given the general trend toward reducing material consumption in machine design, the research on CWR for hollow parts will be continued. Future investigations will focus on manufacturing parts with more complex shapes than those with single reductions in an area.

## 9. CWR Formation of Parts Made of Non-Ferrous Materials

The interest in using CWR to produce parts made of non-ferrous materials results from the need to replace steel with other materials of comparable strength yet lower density. Titanium alloys are the most popular in this respect. Gontarz et al. [83] conducted the first FEM analysis of CWR for producing a Ti6Al4V alloy shaft. Owing to its properties, this titanium grade was used in later studies devoted to the applications for CWR. Cakircali et al. [29] investigated the cross-wedge rolling process conducted at 500 °C and 750 °C. Pater et al. [84] investigated CWR combined with upsetting, finding that the solution made it possible to produce a shaft with its diameter larger than that of the billet. Li et al. [85] examined the effect of the CWR parameters (*α*, *β*, *T*, and Δ*A*) on the shape and dimensions of a rolled part. Ji et al. [86] modeled the CWR of a blade preform. Peng et al. [87] analyzed 46 cases of CWR to determine wear characteristics of the wedge tools. Li et al. [88,89] investigated the CWR of a vehicle arm preform to determine the optimum temperature of the billet. Feng et al. [90,91] investigated the production of a hollow shaft by CWR with a mandrel.

Another group of frequently studied non-ferrous materials includes aluminum alloys. Pater et al. [92] studied the CWR process for producing toothed and screw shafts made of 2816 alloy. Jia et al. [93] investigated the effect of CWR parameters on the internal crack formation in parts made of 7075 alloy. Wang et al. [94,95] investigated the effect of billet temperature in the cross-wedge rolling of 6082 alloy shafts. To obtain high quality products, the tools had to be preheated to 300 °C. Chen et al. [96] modeled the forming of an automobile rear upper control arm from a 6082 alloy preform. The temperature of the tools was maintained constant at 350 °C.

Few studies have been devoted to the problem of rolling parts from other non-ferrous materials. Tomczak et al. [97] simulated the CWR process for producing a lever preform made of AZ31. Mirahmadi et al. [98] conducted the FEM analysis of CWR for a nickel-based superalloy shaft to determine tool angles that would ensure the stability of this rolling process. Lu et al. [99] investigated cold micro-CWR for producing micro-components made of pure copper.

It is believed that more research studies will be conducted in the future on the use of CWR for producing parts made of non-ferrous metals and their alloys. This will result from a trend toward replacing steel parts by their more lightweight counterparts made of non-ferrous metals. Moreover, it is expected that the interest in CWR for producing hybrid parts made of at least two different materials will increase.

## 10. New CWR Methods

FEM has also been used to develop new CWR methods. Pater et al. [100] modeled the cross-wedge rolling process conducted with the use of one flat wedge and two forming rollers. This method was proved viable for manufacturing solid and hollow parts of simple geometry [9]. A different and highly efficient solution is CWR conducted by two rolls with helical wedges. Pater et al. [101] investigated this CWR method to assess its feasibility for producing grinding media balls. Helical-wedge CWR was also studied, in which several balls were formed simultaneously during one revolution of the rolls. This method can also be employed to produce parts such as workholding bolts [102] and ball pins [103].

FEM analyses have also been conducted on multi-wedge CWR, wherein the workpiece is deformed by several pairs of wedges at the same time. This manufacturing method is desirable when many parts of simple geometry are to be formed simultaneously. An example of this method is the CWR of balls by flat wedges, which is described in [104]. Multi-wedge CWR can also be employed to produce long shafts, the manufacture of which—if conducted in a conventional way—would require the use of very long tools. Examples of the FEM modeling of multi-wedge CWR for producing long shafts are given in [74,75,105].

FEM has also been applied to model CWR processes for producing large-sized products, which cannot presently be manufactured under industrial conditions due to a lack of rolling mills of that size. Such products include railcar axles. Numerical results demonstrated that in the classical solution, the spacing between the axes of the mating rolls had to be at least 1800 mm. Force parameters and rolling mill engine power were also determined, showing that the latter could be reduced when CWR was conducted using wedges with convex forming surfaces. The nominal diameter of the rolls could be reduced to 1500 mm if the CWR process for railcar axles was conducted with the use of multi-wedge tools or even to 1200 mm if the parts were rolled in two separate tool passes (with the central step forming in the first pass and the steps on the workpiece ending in the other) [106].

In the future, FEM will remain the basic tool for designing new CWR-based techniques as it is today. This approach will primarily result from the need to reduce costs and time related to the development of new solutions. Consequently, only solutions developed following in-depth numerical analyses will be qualified for experimental validation under industrial conditions.

## 11. Example of CWR Process Modeling

To show the potential of FEM in CWR modeling, the CWR process for a stepped shaft presented in Figure 1 was simulated numerically. This type of shaft is used in car gearboxes and is characterized by a highly complex shape. The largest diameter region of the rolled shaft is 75 mm, which makes it 3.75 times larger than the smallest diameter region of this part (which is 20 mm). Assuming that the billet diameter is equal to the largest diameter of the rolled shaft, this means that the maximum area reduction will be as high as 92.9%. Two passes of the wedge tool will be required to achieve this cross-sectional reduction.

The design of tools for manufacturing a shaft with such complex geometry is difficult and usually carried out in several successive steps, each of which is verified by numerical modeling. Naturally, the number of these steps greatly depends on the designer’s experience and knowledge of the CWR process. The previous method of tool design was expensive because the developed solution had to be validated via industrial rolling tests. As a result, many users of cross-wedge rolling mills commissioned the design and construction of CWR tools to rolling mill manufacturers because these companies employed engineers with considerable experience in CWR. As a result, the implementation costs of these tools were high and the number of new users of CWR was limited. Thanks to FEM, this limitation has been effectively overcome.

Figure 2 shows the wedge roll used in the analyzed CWR process of a stepped shaft. The wedges are wound on the face of the roll having a diameter of 850 mm. The spacing between the axes of the mating rolls is set to 1000 mm. The wedges are described by a forming angle α and a wedge angle β, the values of which are given in the caption. At their ends, the tools have cutters for cutting off excess material on the ends of the workpiece.

The numerical simulation was performed using the commercial software Forge^®^ NxT v4.0. This software was previously used to model CWR processes [28,106,107,108,109,110,111,112,113], and the obtained numerical results showed high agreement with experimental findings.

A geometrical model of the studied CWR process for producing a stepped shaft is shown in Figure 3. The model consists of two identical wedge rolls, two identical guides, and a cylindrical billet with a diameter of 73 mm and a length of 114 mm. The billet was made of 41Cr4 steel, and the material model of this steel grade was taken from the material database library of the Forge^®^ NxT 4.0 simulation software. The temperature of the billet was set to *T* = 1200 °C and the temperature of the tools was maintained constant at *T_T_* = 250 °C. The exchange of heat between the tools and the workpiece was described by a heat exchange coefficient of 10 kW/m^2^ K. Both rolls were rotated in the same direction with a speed of 6 rev/min. Friction on the contact surface was described by the Tresca friction model, with the friction factor set to 1 and 0.6 for the rolls and guides, respectively.

The workpiece was modeled with the use of tetrahedral finite elements. Automatic remeshing occurred when the effective strain increased by a value of 0.4 in any element of the mesh. The discretization of the workpiece and rolled part into the finite elements is shown in Figure 4. Figure 5 shows how the number of nodes and finite elements used for modeling the workpiece changed during the calculations.

The numerical simulation was performed on a 32-core personal computer. The CPU time was 91 h. As it can be observed in Figure 6, the computations were the most time-consuming in the final stage of the CWR process, i.e., when the excess material on the workpiece ends is cut off.

Figure 7 shows how the shape of the workpiece changed in the CWR of a stepped shaft. It can be observed that the forming process is stable and free from any disturbances. When cut off is executed in the final stage of the forming process, the excess material on the workpiece ends keeps the workpiece stable between the guides and prevents it from undesirable skewing. The workpiece material undergoes considerable axial displacement due to the impact of the side walls of the wedge. As shown in Figure 8, the displacement increases with increasing area reduction. The areas where the axial displacement of the material do not occur coincide with the region where the first wedge cuts into the workpiece.

Figure 9 shows the temperature distribution in the rolled part. The temperature of the workpiece material is high despite the fact that the forming time was relatively long. This results from the fact that the heat carried away to the much colder tools is compensated for by the heat generated by the work of friction and plastic deformation. The temperature of the workpiece allows for conducting necessary heat treatments without reheating the workpiece.

In CWR, the material undergoes extensive plastic deformation. This results not only from area reduction, but also from a rapid circumferential flow of the material due to the impact of friction forces. Material torsion causes greater strains in the surface layers, as shown in Figure 10.

The use of FEM makes it possible to predict internal crack formation in the rolled part. Figure 11 shows the distribution of the damage function calculated according to the Cockcroft–Latham ductile fracture criterion. The results demonstrate that the highest values of this function are located in the axial zone of the longer end of the shaft. For fracture to occur, these values (which are about 3) must be greater than the critical damage function value, which depends on the material temperature and is determined via so-called calibration tests. The highest values of the damage function are located where the temperature is about 1140 °C. The specialist literature lacks information on the critical values of the damage function for the 41Cr4 steel grade. For comparative purposes, one may however use the critical values of this function determined for other steel grades. The critical damage function calculated with the Cockcroft–Latham criterion via a CWR test is 3.6 for 42CrMo4 steel formed at *T* = 1138 °C and 4.3 for C45 steel formed at *T* = 1133 °C [40]. It can therefore be assumed that the studied CWR process for producing a stepped shaft should be free from internal crack formation.

Figure 12 shows the predicted wear of the wedge tools. The greatest tool wear is located at the fillet where the forming surface of the tool changes into the sizing surface. This pattern of tool wear should be considered favorable because this region will gradually shift in a direction opposite to that of the rotation of the tools. In contrast, the wear of the tool sizing surface (cylindrical) is minimal, which will ensure high rolling accuracy.

Figure 13 and Figure 14 show the distributions of force parameters in the analyzed CWR process, i.e., radial loads and torques. In terms of quality, these distributions are nearly identical. The highest values of the radial load (which affects rolling accuracy) and torque (which describes rolling mill power) occur at an early stage of the CWR process, when the tools affect the diameter of the workpiece. In later stages of the rolling process, the wedge tools exert impact on the regions of the workpiece that were deformed in the early stage of CWR. The diameter of these regions is smaller than that of the billet, which leads to a considerable decrease in the force parameters. The knowledge of maximum radial loads and torques is of vital importance when it comes to forging mill design.

## 12. Summary

This paper presented a comprehensive overview of the research studies on the application of the finite element method to cross-wedge rolling modeling that were undertaken in the last twenty-five years. Considering the research problems discussed in this work, the following conclusions have been drawn:It is standard practice in numerical modeling of CWR processes to analyze stresses, strains, and temperatures, determine force parameters, as well as predict the formation of external defects on the workpiece, such as necking, bending, overlap, and cross-sectional distortion.Accurate numerical modeling of internal crack formation in the workpiece deformed by CWR, including the prediction of crack location and size, is currently difficult to perform and requires further research work.Given the general trend toward reducing material and energy consumption, new studies will be conducted on the CWR process for producing hollow parts and parts made of lightweight non-ferrous materials. It is also believed that the interest in CWR for producing hybrid parts made of at least two different materials will increase.The prediction of mechanical properties and the microstructure of parts produced by CWR will be carried out on a more extensive basis, with numerical modeling extended to cover both billet preheating and final heat treatment.FEM will remain the basic tool for developing innovative CWR-based manufacturing techniques, including those dedicated to the production of large-sized parts such as railcar axles.The fact that complex CWR processes can be modeled numerically and hence the design of wedge tools is easier will lead to a growing interest in this advanced manufacturing technique.

## Figures and Tables

**Figure 1 materials-16-04518-f001:**
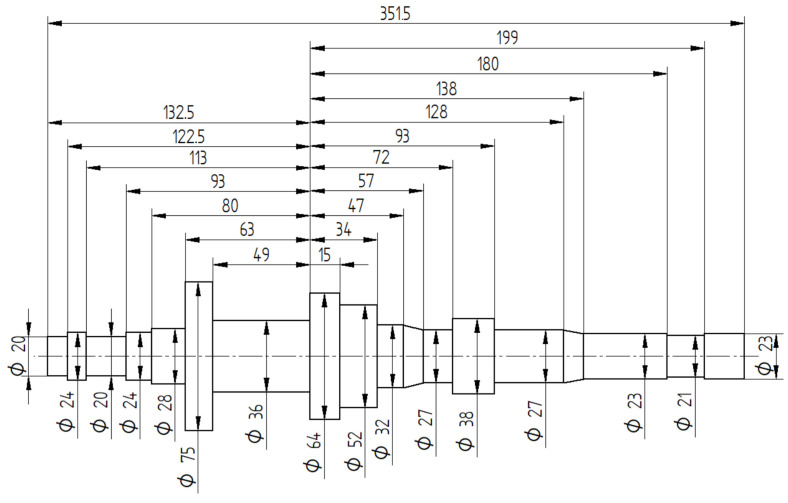
Stepped shaft used in numerical analysis.

**Figure 2 materials-16-04518-f002:**
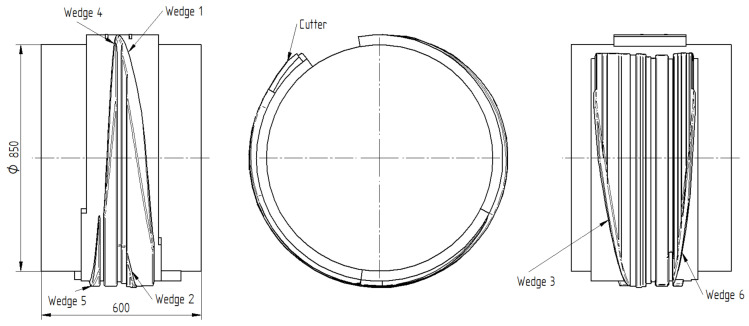
Wedge tool used in FEM analysis of CWR of a stepped shaft; the wedges are described by the following angles: Wedge 1—*α* = 22.5°, *β* = 9.4°; Wedge 2—*α* = 22.5°, *β* = 8.4°; Wedge 3—*α* = 22.5°, *β* = 9.1°; Wedge 4—*α* = 22.5°, *β* = 5.9°; Wedge 5—*α* = 22.5°; *β* = 8.3°; Wedge 6—*α* = 22.5°, *β* = 6.6°.

**Figure 3 materials-16-04518-f003:**
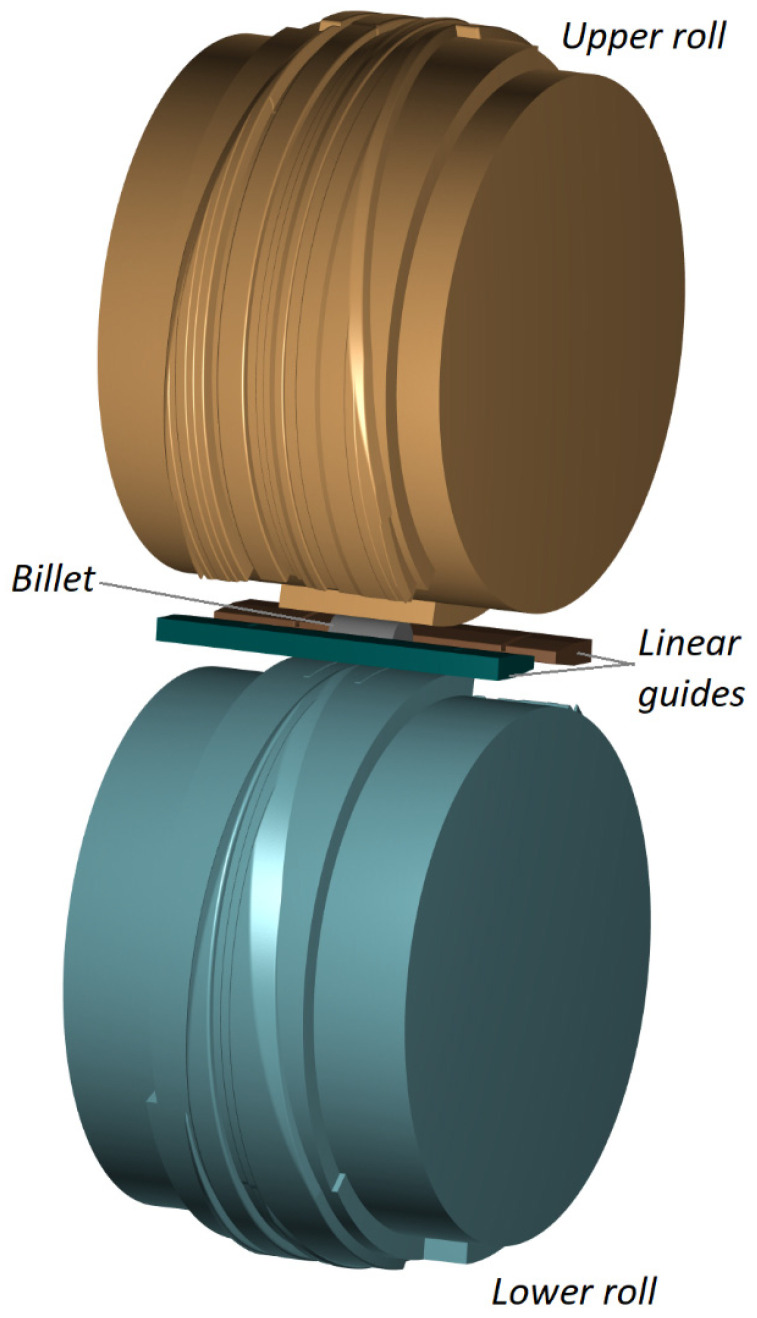
Geometrical model of a CWR process for producing a stepped shaft.

**Figure 4 materials-16-04518-f004:**
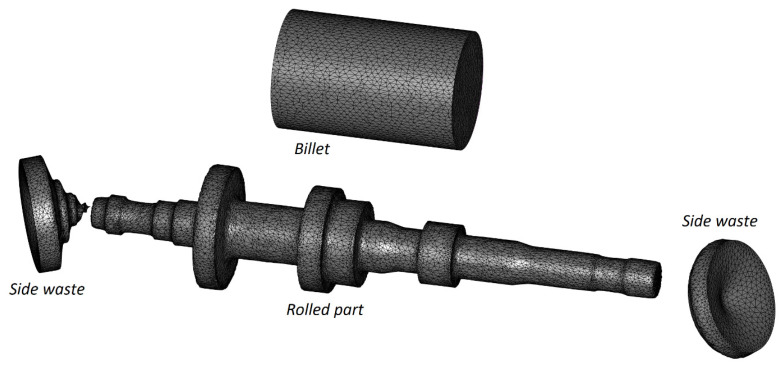
Division of the billet and rolled part with excess material on its ends into tetragonal finite elements.

**Figure 5 materials-16-04518-f005:**
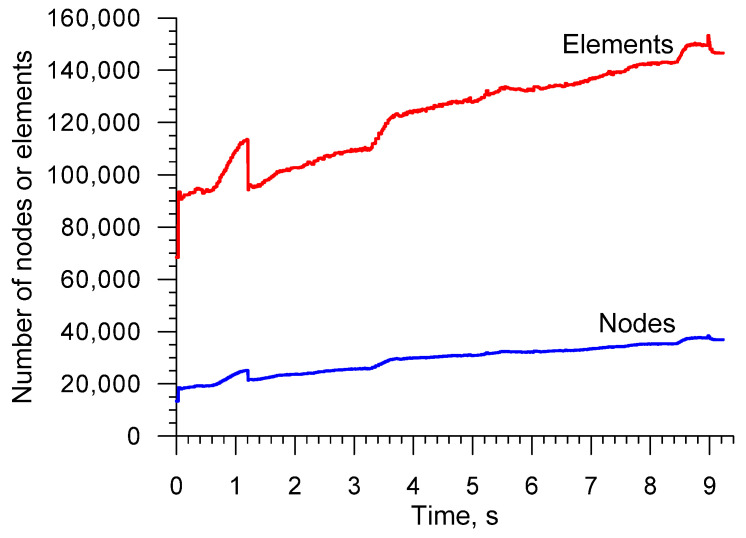
Number of nodes and elements that were used to model the workpiece in the CWR process under study.

**Figure 6 materials-16-04518-f006:**
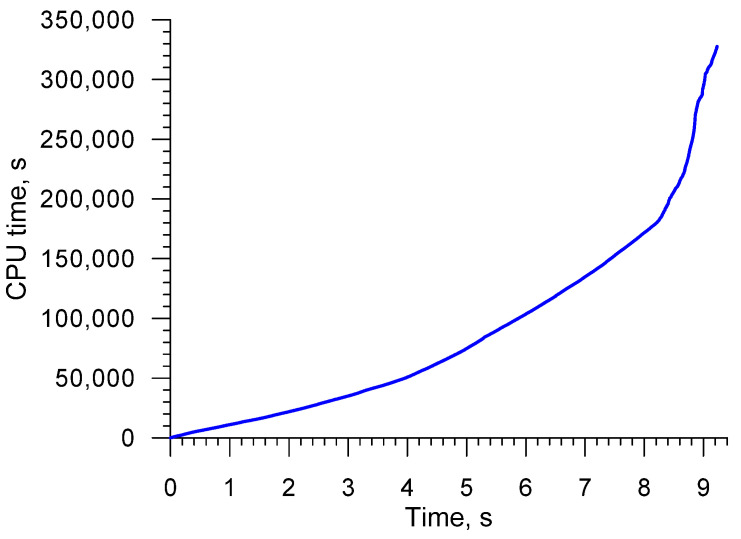
CPU time vs. time of CWR process.

**Figure 7 materials-16-04518-f007:**
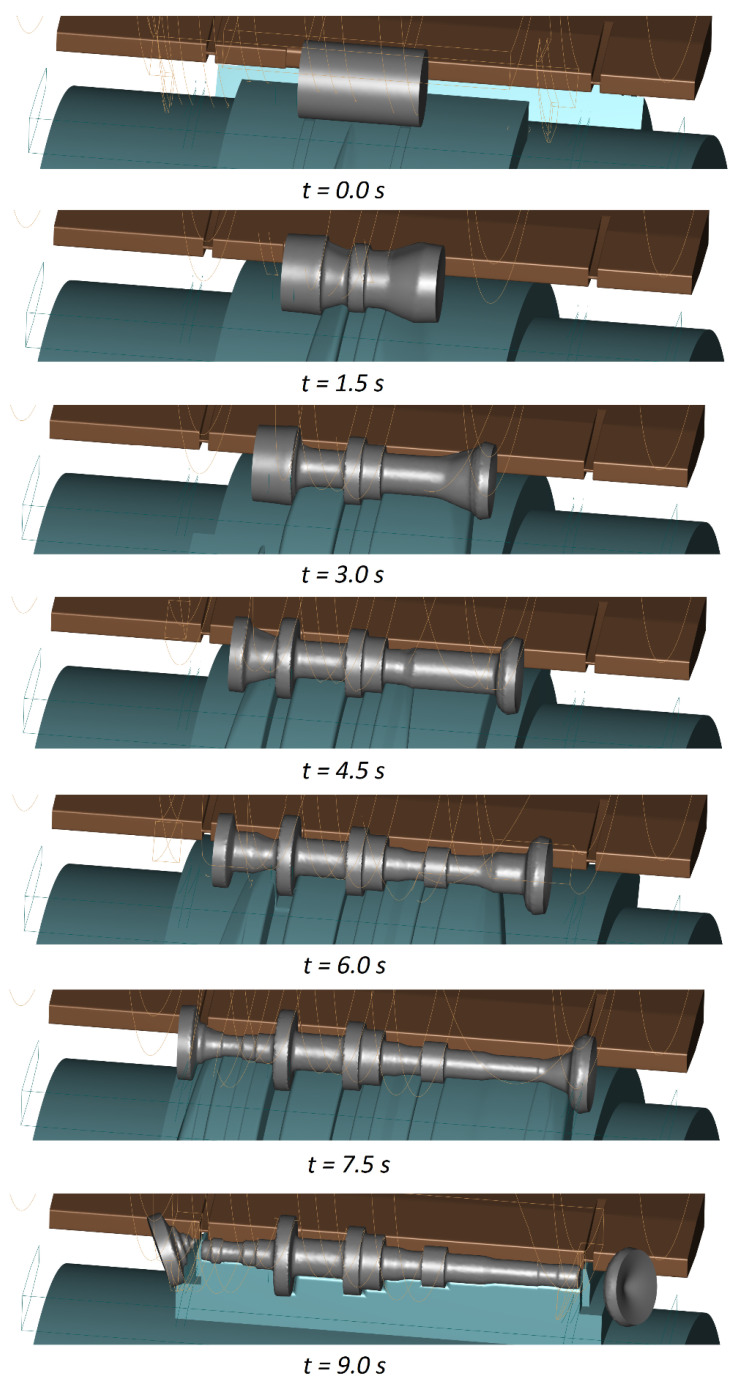
Changes in the shape of the workpiece in CWR of a stepped shaft (the tools are colour-coded as shown in Figure 3).

**Figure 8 materials-16-04518-f008:**
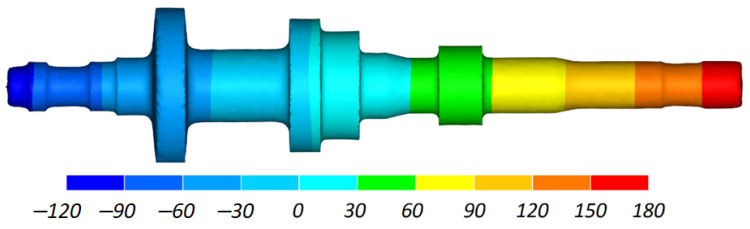
Material axial displacement (in mm) in a stepped shaft produced by CWR.

**Figure 9 materials-16-04518-f009:**
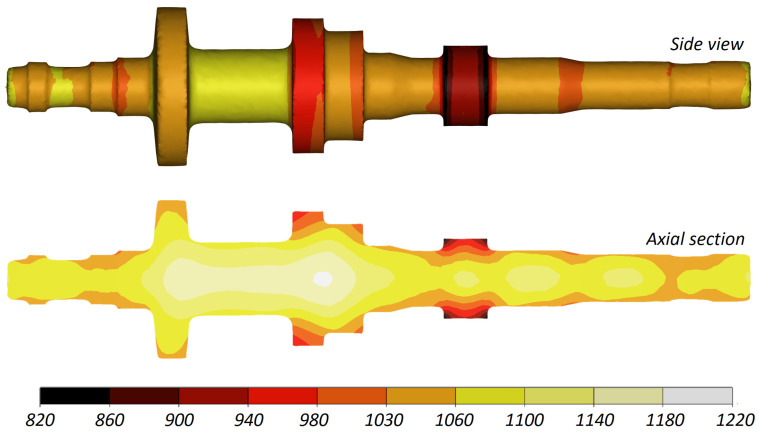
Temperature distribution (in °C) in a stepped shaft produced by CWR.

**Figure 10 materials-16-04518-f010:**
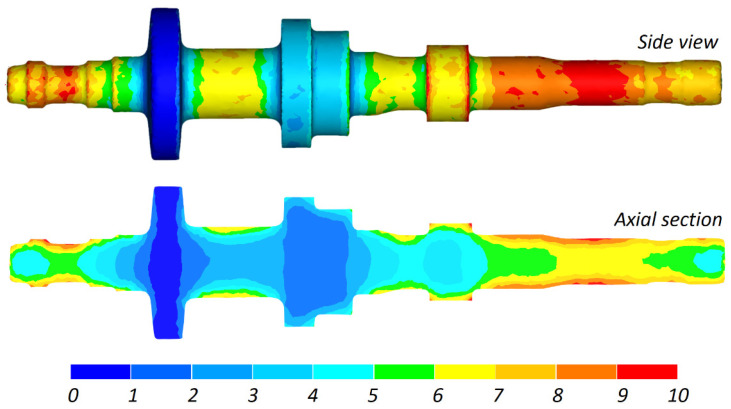
Effective strain distribution in a stepped shaft produced by CWR.

**Figure 11 materials-16-04518-f011:**
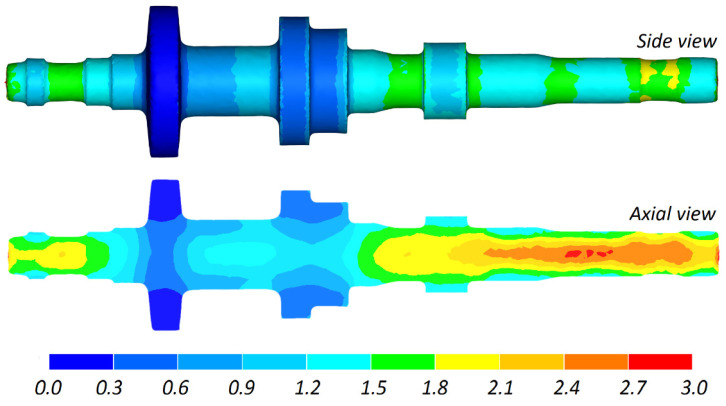
Damage function distribution (calculated according to the Cockcroft–Latham criterion) in a stepped shaft produced by CWR.

**Figure 12 materials-16-04518-f012:**
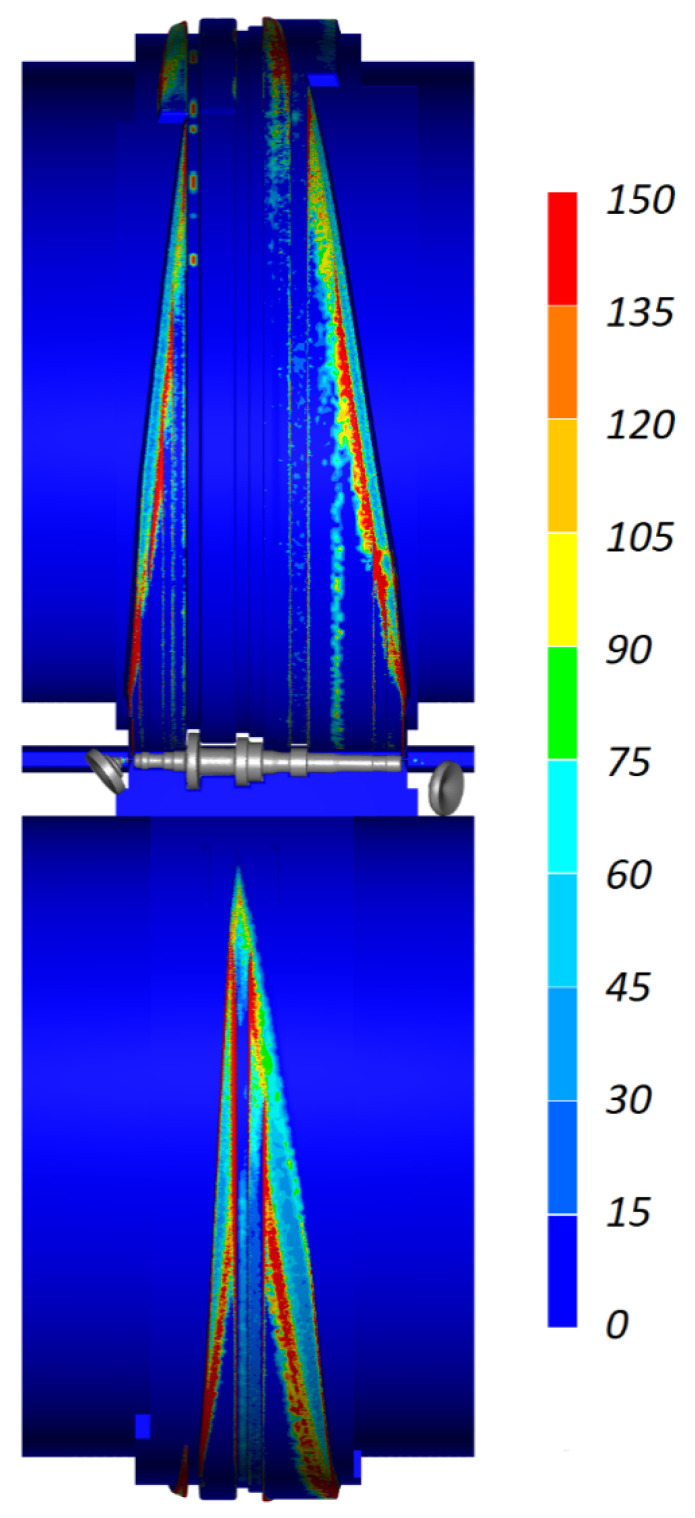
Wedge tool wear (expressed in mm·MPa) in CWR of a stepped shaft.

**Figure 13 materials-16-04518-f013:**
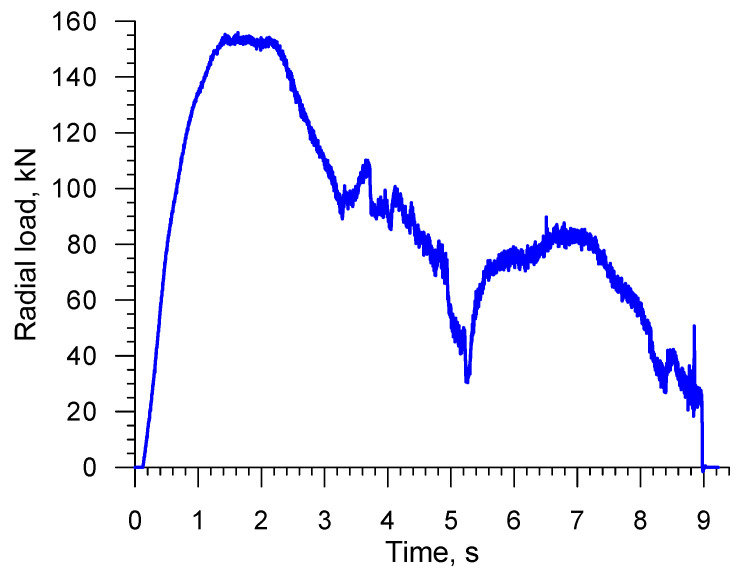
Radial load distribution in CWR of a stepped shaft.

**Figure 14 materials-16-04518-f014:**
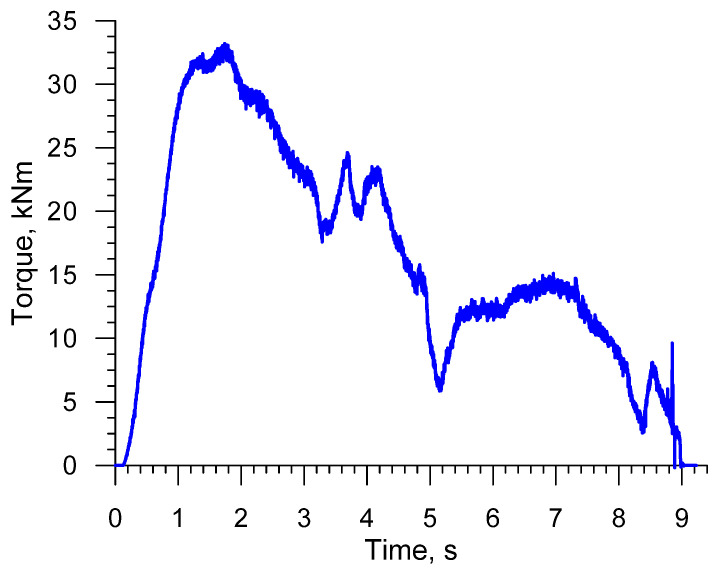
Torque distribution in CWR of a stepped shaft.

## Data Availability

Data sharing not applicable.

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
