# Peer review of "The Application of Finite Element Method for Analysis of Cross-Wedge Rolling Processes—A Review"

_materials, 2023, doi:10.3390/ma16134518_

Round 1

Reviewer 1 Report

The presented work emphasized an extensive review on the applications of finite element analysis on cross wedge rolling process. The reviewed work and literatures were appropriate, however, following shortcomings are presented in the manuscript and which has to be rectified before further processing. I request mandatory revision, as listed below, please do not simply respond but revise manuscript.

·       Since, the cross-wedge rolling is a well-established process and used by many industries to fabricate axles and shafts. Therefore, it is necessitated to describe the usefulness of FEA models for such well-established process in the problem definition.

·       If the figures used in this manuscript were reproduced from the existing works, it is mandatory to cite the references appropriately.

·       Conclusion (Summary) of this manuscript is not effectively describing the outcome of the article.  It is suggested to highlight the limitations of this study, suggested improvements to this work and future directions in the conclusion section. Also, the conclusion can be presented better than the present form with more findings.

·       The organization of work is not adequately described. The sub headings and main topics should be described appropriately as per the review article formats.

·       Finally, the similarity content is found to be 29% (Turnitin). It should be minimized.

Reviewer 2 Report

1.- The abstract lacks structure and does not show the significance of the study. A background is also needed for the abstract.

2.- The conclusions are elementary and do not highlight anything of the research conducted; they should be reformulated. 

3.- The case study ( Figure 1. Stepped shaft used in numerical analysis) you propose should be complemented better; for example, you can check the following reference https://link.springer.com/article/10.1007/s00170-021-07678-z. I have the impression that the manuscript looks like a good laboratory practice.

A native English speaker should check your manuscript's grammar, spelling, punctuation, and phrasing to improve its readability.

Reviewer 3 Report

The paper presents a review study on the application of FEM to analysis of cross wedge rolling processes. Different research areas such as stress analysis, damage mechanisms, microstructural evolution etc. are discussed. I think this study is interesting and good for the field. I suggest publication after addressing the points given below.

·        The text should be checked for typos and grammar.

·        A nomenclature section may be given.

·        I think the sections gives an insight about the topic but not reflecting the issue in detail. Author may extend the discussions pointing out the key features.

·        The paper should include more figures to support the texts.

·        Section 6 and 7 have the same heading as “Microstructure modelling”.

·        Conclusions should be extended.

Mostly fine. 

Reviewer 4 Report

The aim of this review article is to familiarize readers with the application of the finite element method (FEM) to cross wedge rolling (CWR) modelling. Several areas in which FEM is applied are described in the paper, namely: stress and strain states, force parameters, CWR failures, material fracture, microstructure modelling, etc. Finally, the process of numerical modelling of CWR in the fabrication of a stepped shaft used in automotive transmissions is shown. As modelling of cross wedge rolling by finite element method has its own specificities, the paper may be of interest to researchers involved in modelling of manufacturing technology processes. There are a few minor formal errors in the paper, which are indicated in the attached file. These are mainly the different form of journal articles notation in the References section.

Round 2

Reviewer 1 Report

The manuscript can be accepted for the publication.

Author Response

Thank you very much for reviewing my article.

Reviewer 2 Report

1. It would be best if we modified Figure 2 a bit. It seems unclear and could confuse the reader. The colors used in the figure also don't help aid understanding.

2. Please include units from Equations (1)-(4). 

3. Figure 3, which is very difficult to understand due to the colors used, should be modified.

4. Change the colors used in Figure 7 because they do not contrast to visualize better the idea you want to convey. 

5. Hey there, I noticed that in Figure 9, the Axial Section is a bit difficult to see. Is it possible to change the color to make it more visible? Thank you!

6. Regarding Figure 12, the stepped axis can be difficult to see.  Please modify the image; this will help the reader understand the idea you want to convey. 

7. Add more details of your work in the conclusions, please. They are constrained. 

 English very difficult to understand/incomprehensible

Reviewer 3 Report

Revised version is acceptable. 

Author Response

(The authors gave the same response as above.)
